# Hyperspectral Target Detection Methods Based on Statistical Information: The Key Problems and the Corresponding Strategies

**Luyan Ji [1,2,†] and Xiurui Geng [1,2,3,*,†]**

1    Aerospace Information Research Institute, Chinese Academy of Sciences, Beijing 100094, China;
     jily@mail.ustc.edu.cn
2    The Key Laboratory of Technology in Geo-Spatial Information Processing and Application System,
     Chinese Academy of Sciences, Beijing 100190, China
3    The School of Electronic, Electrical and Communication Engineering, University of Chinese Academy of
     Sciences, Beijing 100049, China
*    Correspondence: gengxr@aircas.ac.cn
†    These authors contributed equally to this work.

**Abstract:** Target detection is an important area in the applications of hyperspectral remote sensing. Due to the full use of information of the target and background, target detection algorithms based on the statistical characteristics of an image are always occupy a dominant position in the field of hyperspectral target detection. From the perspective of statistical information, we firstly presented detailed discussions on the key factors affecting the target detection results, including data origin, target size, spectral variability of target, and the number of bands. Further, we gave the corresponding strategies for several common situations in the practical target detection applications.

**Keywords:** hyperspectral; target detection; target size; spectral variability; band selection; CEM

## 1. Introduction

Hyperspectral remote sensing has developed rapidly in the past few decades (Figure 1a) and plays a crucial role in the field of remote sensing. Compared with traditional optical data, hyperspectral images contain rich spatial and spectral information of ground objects, which can provide sufficient data for effective extraction of various land covers and high-precision quantitative analysis [1]. Among all the land cover mapping techniques for hyperspectral images, target detection has always been an important research area [2]. Theoretically, target detection in hyperspectral remote sensing is a binary problem [3], whose goal is to find an effective detector that can enhance target pixels while suppress background pixels as much as possible.

Existing target detection algorithms for hyperspectral images can be roughly divided into the following categories:

(1)    Target detection based on spectral characteristics. When the target of interest has diagnostic absorption features, they can be detected directly based on these spectral bands or features [4]. Commonly used quantitative parameters for absorption-band features include the position, depth, width and asymmetry [5,6]. These parameters are widely used for determining the mineralogy of samples or analyze the chemical composition [7,8]. To eliminate the effects of atmosphere, lighting differences, and other factors, the continuum is often removed before spectral feature extraction [9]. Moreover, continuum removed spectra can highlight the absorption characteristics. In addition, spectral indices based on target spectral characteristics can also be designed for target detection, such as the normalized difference vegetation index (NDVI), soil-adjusted vegetation index, enhanced vegetation index for vegetation,

normalized difference water index, and modified normalized difference water index for water [10–16]. These methods are developed by utilizing the spectral characteristics of targets, so they are physical-based. Generally, they are simple with low computational complexity. However, they typically only utilize a small portion of bands (usually bands with absorption features), and completely ignore background information, so their accuracy is often lower than needed.

(2) Target detection based on spectral similarity matching. These methods aim to measure the spectral similarity of the target and each pixel in the image, so as to directly obtain the distribution of target in the image. Common methods include minimum distance method [17], spectral angle mapper [18,19], cross correlogram spectral matching [20], spectral similarity scale [21], and correlation simulating analysis model [22]. This kind of method is also simple and easy to implement, but they have not used background information either. As a result, their accuracy can not be well guaranteed in many cases.

(3) Target detection based on spectral mixture analysis. This kind of method relies on the assumption of a linear mixing model, which means that each pixel in the image can be a linear combination of several pure spectral signatures (also named endmembers) weighted by the corresponding abundance fractions. Owing to the physical constraints, the abundances satisfy the abundance non-negative constraint and abundance sum-to-one constraint [23]. The commonly used procedure to extract the distribution of target is as follows: (a) extract endmembers from the image through endmember extraction methods [24,25]; (b) identify each endmember as target or background; and (c) perform spectral unmixing using the target endmember and background endmenbers [26]. Then the abundance map of the target endmember is the distribution of target. In addition, a simple spectral matching can also be used to detect target in the orthogonal subspace of the background endmembers. The most representative way to calculate the orthogonal subspace is to perform the orthogonal subspace projection [27]. Based on this algorithm, noise subspace projection [28], generalized orthogonal subspace projection [29], have also been developed. The above methods are all based on the assumption that there are pure pixels of all kinds of objects in the image. However, in real images, this assumption often does not hold. To address this issue, researchers have proposed endmember generation methods that can simultaneously obtain endmembers and abundance maps by proposing a reasonable objective function [30–32]. The algorithm based on spectral mixture analysis can effectively extract targets and suppress background in many cases, but its detection accuracy is easily to be affected by the errors in background endmember extraction. In addition, the linear mixing model is not always valid in some scenarios [33,34].

(4) Target detection based on image statistical information. This type of methods relies mainly on the statistical information of the hyperspectral data. Its aim is to suppress the background while matching the target. The most classical algorithms in this category are the Constrained Energy Minimization (CEM) algorithm [35] and the Matched Filter (MF) algorithm [36,37]. The former was proposed by Harsanyi in 1993, which was derived from a linearly constrained adaptive beamforming in the field of signal processing. CEM aims to minimize the average filter output energy while maintaining the output for the target as a constant. The latter is derived from the binary hypothesis testing problem in statistics, assuming that both the target and the background classes follow the Gaussian distribution. MF obtains the optimal target detector by maximizing the likelihood ratio under the Neyman–Pearson criterion. The detectors of MF and CEM have similar mathematical expressions, with the only difference being that MF requires the data to be centralized first. Many algorithms have been developed based on these two algorithms, such as the Mixed Tuned Matched Filter [38], General Constrained Energy Minimization [39], Adaptive Cosine Estimation [40], Multi-Objective Constrained Energy Minimiza-

tion [41], Target-Constrained Interference Minimization Filtering [42], Kernel-based Constrained Energy Minimization/Matched Filtering [43], Weighted Constrained Energy Minimization [44], Clever Eye Algorithm [45], Augmented Constrained Energy Minimization [46], and Filtering Tensor Analysis [47,48]. In addition to second-order statistical quantities, higher-order statistics have also been introduced for hyperspectral target detection [49–52]. By making full use of the statistical information of the background, this kind of algorithms can achieve good target detection results even when the background spectrum is unknown. However, if there exist a large spectral variation in target pixels or the target's spectra are close to that of the background, this kind of methods may fail to achieve a high target detection accuracy.

(5) Target detection based on deep learning. In recent years, due to the strong data analysis capabilities, deep learning has been introduced to various hyperspectral applications [53–59], especially for classification [60–62]. Yet, unlike hyperspectral classification, there has been relatively less research on deep learning-based hyperspectral target detection algorithms [63,64]. There are two types of deep learning-based methods for hyperspectral target detection [65]. One way is to treat the task as a deep learning-based binary classification problem, by setting the target pixels as positive samples and the background pixels as negative samples [63,64,66,67]. Due to the the limited sample problem, this kind of methods usually require to perform training sample augmentation [63,64,68]. Another way is to use the deep learning model by combing the feature reconstruction and target detection together [69,70], and the generative models includes autoencoder [69], variational autoencoder [71], and generative adversarial networks [72]. Compared to the traditional methods, deep learning methods typically achieve a higher accuracy, yet they often lack interpretability. In addition, they usually require a higher computational complexity and more parameter tuning.

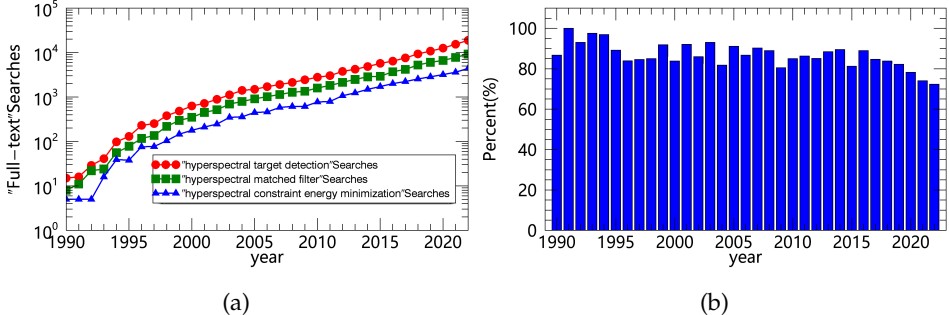

(a)　　　　　　　　　　　　　　　　　　　　　　(b)

**Figure 1.** The results obtained by searching "hyperspectral target detection", "hyperspectral constraint energy minimization", and "hyperspectral matched filter" on Google Scholar via "Full-text" Searches since 1990 (**a**), and the percentage of the "hyperspectral constraint energy minimization" search number + "hyperspectral matched filter" search number to the "hyperspectral target detection" search number (**b**).

From Figure 1b, we can see that their percentage has slightly declined since 2019, CEM and MF are always playing an important role in the field of hyperspectral target detection. Moreover, the commonly used remote sensing software, Environment for Visualizing Images (ENVI, https://www.nv5geospatialsoftware.com/Products/ENVI/ (accessed on 30 July 2023).) also includes the two methods for users. In addition, compared to deep-learning based methods, since the mathematical expressions of CEM and MF detectors are clear and precise, it is easy to explain the phenomena that occur in hyperspectral target detection in mathematical way. Therefore, in this paper, based on these two methods, firstly we will theoretically explore the factors that impact the performance of target detection. Secondly, we will provide corresponding strategies under four typical scenarios. The main contributions of this work as summarized as follows:

- We have theoretically analyzed the factors that affect the target detection, including the data origin, target size, target spectral variation, and the number of bands.
- We have theoretically proven that the impact of the target size mainly depends on the degree of spectral variation in the target. If all the target pixels have identical spectra, this factor will not impact the final output.
- Increase the number of bands can improve the performance of the detector from the perspective of average filter output energy. However, when there is significant variation in target's spectra, an excessive number of bands will cause the detector to suppress other target pixels as well.
- In four different target extraction scenarios with varying levels of difficulty, the effects of selecting different target spectra, adding or removing bands, using single-target or multi-target detection algorithms, as well as the role of spatial filtering in target detection are demonstrated using multiple real hyperspectral datasets.

## 2. Problems in Hyperspectral Target Detection

The main objective of hyperspectral target detection is to search the pixels of an hyperspectral data for the presence of the target [37,73]. Hyperspectral target detection algorithms based on statistical information typically start by searching for an optimal projection direction (denoted as **w**). The data are then projected onto **w** to produce an output (denoted as **y**). Finally, a thresholding method can be used to generate a target distribution map of the target, as shown in Figure 2.

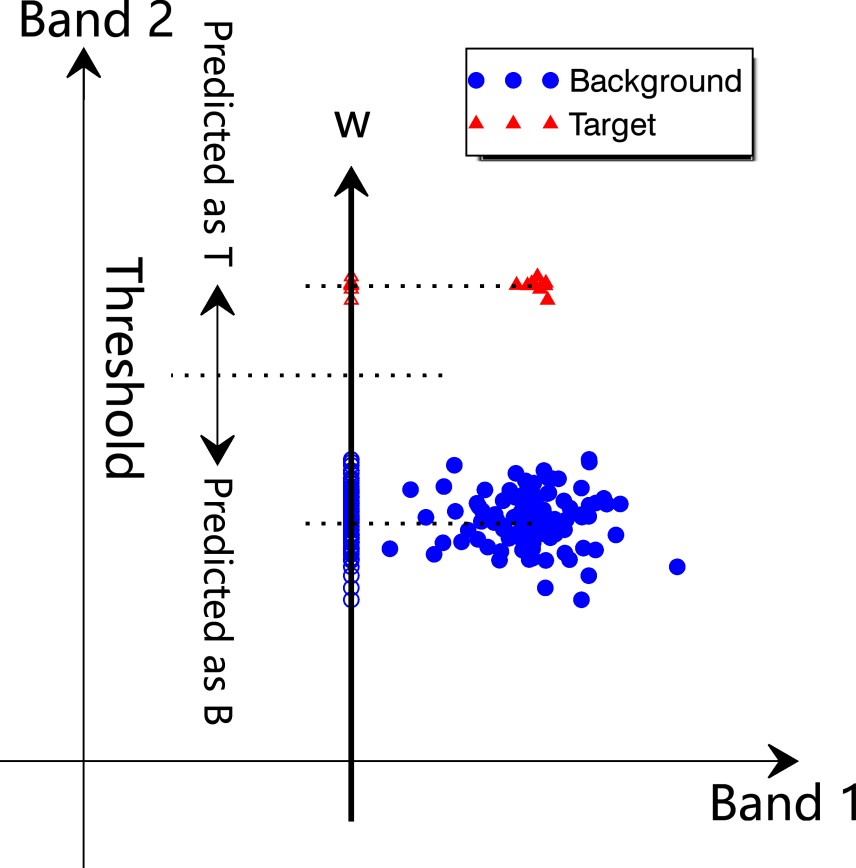

**Figure 2.** Illustration of hyperspectral target detection in the 2-D space. T: target, B: background.

Suppose a hyperspectral image $\mathbf{X} = [\mathbf{x}_1\ \mathbf{x}_2\ \ldots\ \mathbf{x}_N]$ is an $L \times N$ matrix, where $\mathbf{x}_i (i = 1, 2, \ldots, N)$ represents the $i$th pixel, $L$ is the number of bands, and $N$ represents the number of pixels. Suppose there are $N_{\mathrm{t}}(N_{\mathrm{t}} \geq 1)$ target pixels in the image, and generally, one of them is selected as the representative spectrum for target detection, noted as **d**. In

this section, we will theoretically explore the impact of four factors (i.e., data origin, target size, target spectral variation, and the number of bands) on the target detection.

*2.1. Data Origin*

The expressions for the CEM and MF detectors are as follows [35,37]

$$\mathbf{w}_{\text{CEM}} = c_{\text{CEM}}\mathbf{R}^{-1}\mathbf{d} \tag{1}$$

$$\mathbf{w}_{\text{MF}} = c_{\text{MF}}\mathbf{K}^{-1}(\mathbf{d} - \mathbf{m}) \tag{2}$$

where $c_{\text{CEM}} = \frac{1}{\mathbf{d}^{\text{T}}\mathbf{R}^{-1}\mathbf{d}}$, $c_{\text{MF}} = \frac{1}{(\mathbf{d}-\mathbf{m})^{\text{T}}\mathbf{K}^{-1}(\mathbf{d}-\mathbf{m})}$ are constants, $\mathbf{d}$ is the spectral signature of the target, $\mathbf{m}$ is the mean vector of the data, $\mathbf{R} = \frac{1}{N}\sum_{i=1}^{N}\mathbf{x}_i\mathbf{x}_i^{\text{T}}$ is the auto-correlation matrix, and $\mathbf{K} = \frac{1}{N}\sum_{i=1}^{N}(\mathbf{x}_i - \mathbf{m})(\mathbf{x}_i - \mathbf{m})^{\text{T}}$ is the covariance matrix. It can be seen that the mathematical forms of these two detectors are very similar, with the only difference being that CEM uses the zero vector as the data origin, while MF uses the mean vector as the data origin. Usually, we find that the detection results of these two detectors are different, suggesting that the data origin can affect the target detection result. In order to find the optimal data origin, the Clever Eye (CE) algorithm was proposed [45], which introduces the data origin, $\mathbf{\mu}$ as a variable in the objective function as

$$\begin{cases} \min_{\mathbf{w},\mathbf{\mu}} \mathbf{w}^{\text{T}}\mathbf{R}_{\mathbf{\mu}}\mathbf{w} \\ (\mathbf{d} - \mathbf{\mu})^{\text{T}}\mathbf{w} = 1 \end{cases} \tag{3}$$

where $\mathbf{R}_{\mathbf{\mu}} = \frac{1}{N}\sum_{i=1}^{N}(\mathbf{r}_i - \mathbf{\mu})(\mathbf{r}_i - \mathbf{\mu})^{\text{T}} = \mathbf{K} + (\mathbf{m} - \mathbf{\mu})(\mathbf{m} - \mathbf{\mu})^{\text{T}}$ is the auto-correlation matrix of the data when moved to the data origin $\mathbf{\mu}$. After a series of mathematical derivations, the solution to this optimization problem can be solved using the gradient ascent method. However, since the objective function is non-convex, only local optimal solutions, called Clever Eye points, can be obtained.

Figure 3 shows the target detection results of CEM, MF, and CE based on a simulated data, which has a size of 21 pixels × 21 pixels × 2 bands, and a target with a size of 3 pixels × 3 pixels located in the center. The background pixels are generated by the "randn" function in MATLAB, setting the mean value set to 2 for both bands, and standard deviation (STD) set to 1 and 0.25 for Band 1 and 2, respectively, while the target spectral values are all set to $[2\ 4]^{\text{T}}$. The distribution of the simulated data in the 2-D spectral space is shown in Figure 3b. Using the upper left target pixel as $\mathbf{d}$, the target detection results of CEM and MF are shown in Figure 3c,d, and the result of CE using zero vector and mean vector as the initial value for $\mathbf{\mu}$ are shown in Figure 3g,h. And the corresponding average filter output energy against the number of iterations are plotted in Figure 3g,h. Since CEM uses the zero vector as the data origin and MF uses the mean vector as the data origin, the energy values of the two curves in Figure 3g,h when the iteration number =1 are actually the average filter output energies of CEM and MF. Therefore, CE can achieve a lower average filter output energy than both CEM and MF. Moreover, from the perspective of visual interpretation, the CE detector outperforms the CEM detector in suppressing the background (Figure 3c–f).

Furthermore, from Figure 3, it can be observed that the target detection results and final average filter output energy of CE with the zero and mean vector as the initial values are exactly the same. This raises the following questions: will different initial values lead to different CE points? If so, is there an optimal CE point? If not, what is the relationship between these CE points which correspond to local extremes? The answers to these questions are given in the following two theorems.

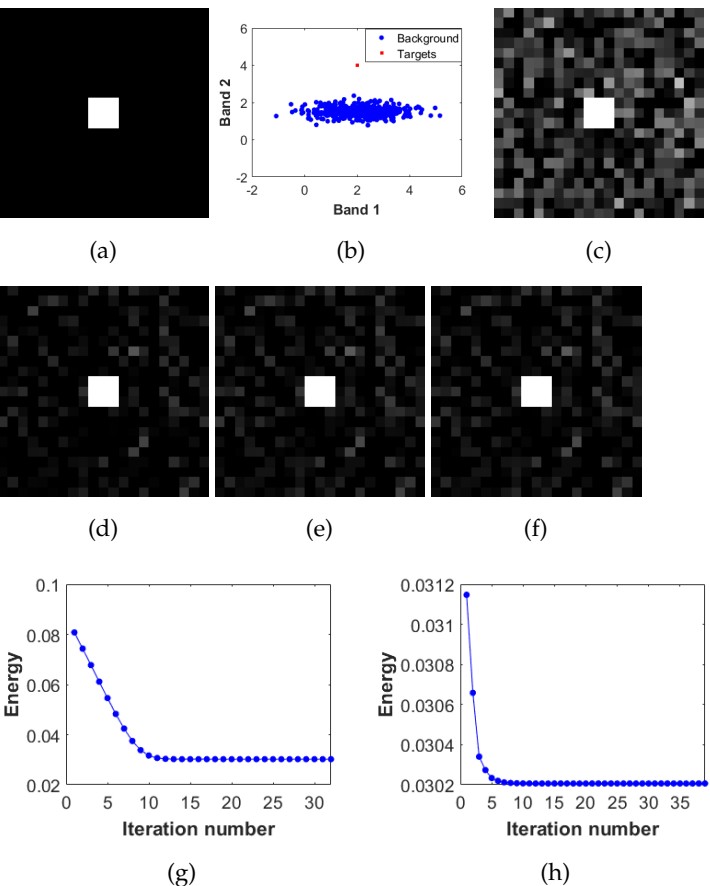

**Figure 3.** Impact of data origin to the result of the target detection. (**a**) Ground truth. (**b**) The distribution of the simulated data in the 2-D space. (**c**) CEM. (**d**) MF. (**e**) CE initialized by a zero vector. (**f**) CE initialized by a mean vector. (**g**) The average filter output energy vs. the iteration number when CE is initialized by a zero vector. (**h**) The average filter output energy vs. the iteration number when CE is initialized by a mean vector.

**Theorem 1.** *All clever eyes satisfy the following linear equation*

$$(\mathbf{d} - \mathbf{m})^{\mathrm{T}} \mathbf{K}^{-1} (\mathbf{m} - \boldsymbol{\mu}) = 1 \tag{4}$$

*which is named the CE equation.*

Please refer to reference [45] for the proof of Theorem 1. In fact, if $\mathbf{m}, \mathbf{d}$ and $\mathbf{K}$ are fixed, the CE Equation (4) actually corresponds to an under-determined linear equation with $L - 1$ degrees of freedom. Theorem 1 indicates that the optimization problem (3) has analytical solutions, which can be directly obtained by solving the linear Equation (4). Figure 4a shows the movement paths of $\boldsymbol{\mu}$ under different initial values using the gradient ascent method on the same simulated data used in Figure 3. It can be seen that CE points calculated by the gradient ascent method are all located on the line of the CE Equation (4).

**Theorem 2.** *For any $\boldsymbol{\mu}$ that satisfies the CE Equation (4), the corresponding detector and filter output energy are both $\boldsymbol{\mu}$-irrelevant, which are*

$$\mathbf{w}_{\boldsymbol{\mu}} = c_{\boldsymbol{\mu}} (\mathbf{d} - \mathbf{m}) \tag{5}$$

*where $c_{\boldsymbol{\mu}} = \frac{1}{(\mathbf{d}-\mathbf{m})^{\mathrm{T}}\mathbf{K}^{-1}(\mathbf{d}-\mathbf{m})+1}$ is a constant independent of $\boldsymbol{\mu}$.*

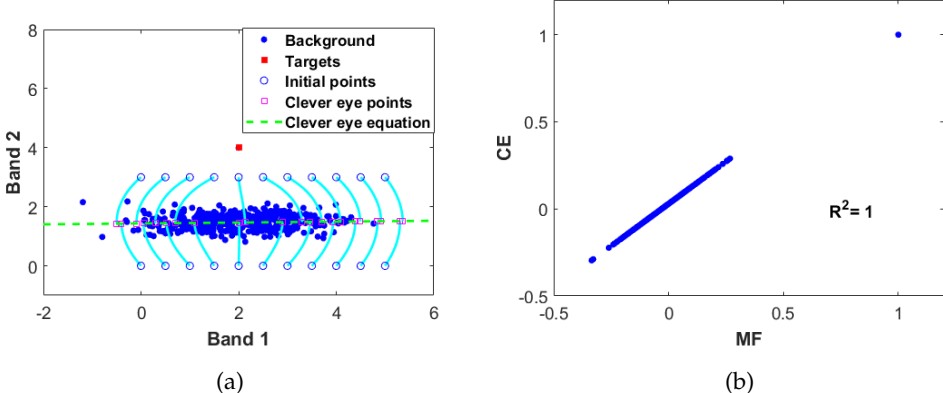

(a)                                                           (b)

**Figure 4.** Results of the CE detector for the simulated data in Figure 3. (**a**) Movement paths of **μ** with different initial points. (**b**) Correlation between the MF's and CE's output.

Please also refer to reference [74] for the proof of Theorem 2. Theorem 2 further proves that all CE points (although they are local optimal solutions) are globally optimal, and correspond to the same projection direction, which means that target detection results of all CE points are the same. Furthermore, comparing the mathematical expression of CE and MF detector (see (5) and (2)), it can be found that the projection directions of the two detectors are identical, which indicates that the target detection results of CE and MF are equivalent. Figure 4b shows the scatter plot of the output results of MF and CE, and we can see that their results are distributed along a straight line with the correlation coefficient equals to 1, which further indicates that the two algorithms are completely equivalent.

On the other hand, from the perspective of energy, the average filter output energy of CE is always lower than that of CEM. Since CE is equivalent to MF, we can conclude that for the two classic hyperspectral target detection algorithms, CEM and MF, the latter is always theoretically superior to the former [46]. Therefore, in hyperspectral target detection, the data should be centered firstly by moving the data origin to the mean vector. Under this condition, the three algorithms, CEM, MF, and CE are completely equivalent. Therefore, the following discussion takes CEM as the representative.

### 2.2. Target Size

In this paper, target size refers to the number of target pixels in the image, i.e., $N_t$ ($N_t \geq 1$ ). However, when using CEM for target detection, only one target spectrum is selected as the representative one for **d**. Previous studies have shown [44] that the larger $N_t$ is, the worse the detection performance of CEM is. Here we use a simulated data to illustrate this phenomenon. Suppose the simulated data have a size of $21 \times 21$ with 100 bands, all of which are generated by the function "randn" in Matlab by setting the mean value to 0 and STD to 1. The target is located in the middle of the image with a size of $n \times n$ ($3 \leq n \leq 19$ , where $n$ is an odd number), which is also generated by the "randn" function by setting the mean value to 10 and the standard deviation to 1 for all bands. Figure 5 shows the ground truth and the data distribution with $n = 9$. Without the loss of generality, we take the upper left corner pixel of the target region as **d**. In order to quantitatively evaluate the detection result, we calculate the area under the receiver operating characteristic (ROC) curve (AUC) of the detection results. Generally, a larger AUC value means a better detection result. Table 1 shows the AUC value as the target size increases from $3 \times 3$ to $19 \times 19$. It can be seen that the detection accuracy does decrease gradually as the target size increases, which is consistent with previous studies.

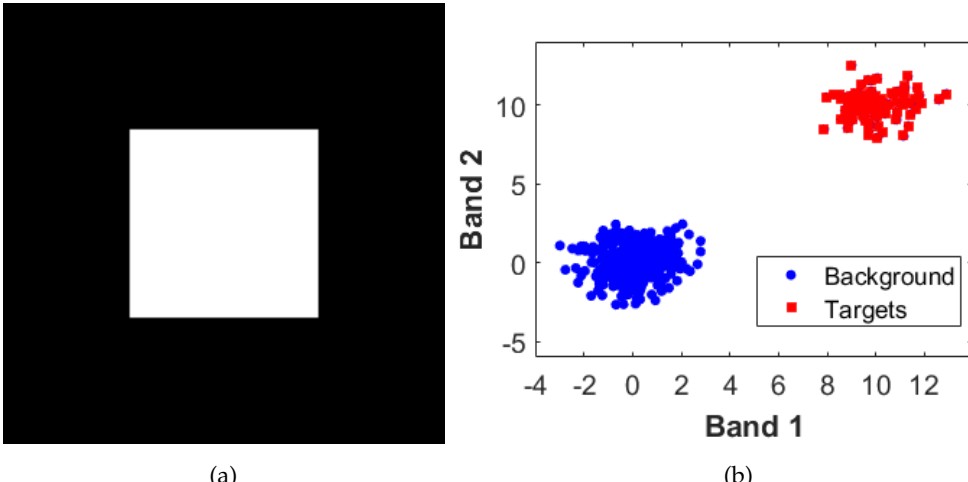

(a)         (b)

**Figure 5.** The simulated data with different target's spectra. (**a**) The ground truth with the target size of $9 \times 9$ and (**b**) the distribution of the data in 2-D feature space.

**Table 1.** AUC values (average of 10 runs) vs. target size for the simulated data in Figure 5.

| Target Size | $3 \times 3$ | $5 \times 5$ | $7 \times 7$ | $9 \times 9$ | $11 \times 11$ |
|---|---|---|---|---|---|
| AUC | 0.9996 | 0.8857 | 0.7250 | 0.6301 | 0.5940 |
| **Target size** | $13 \times 13$ | $15 \times 15$ | $17 \times 17$ | $19 \times 19$ | |
| AUC | 0.5669 | 0.5580 | 0.5369 | 0.5250 | |

The reason for this phenomenon can be explained as follows: taking CEM as an example, the auto-correlation matrix **R** in (1) is composed of all pixels in the image, including the target pixels. This means that the target pixels are not only taken as target signals, but also as part of the background signals. Therefore, when the target size $N_t$ is small, the influence of the target pixels on **R** is neglectable. However, as the increase in $N_t$, the influence cannot be ignored any more. If $N_t$ is large enough, for example, if $N_t$ is comparable to $N$, the CEM detector is no longer able to suppress the background when calculating **R** using all pixels [44].

*2.3. Target Spectral Variation*

In the previous section, we mainly discuss the impact of target size on the performance of target detection, but ignore the influence of spectral variation in target pixels on the detection result. Therefore, in this section, we will focus on how the target detector performs when the target's spectra vary.

Taking the simulated data in Figure 5 as an example, we fix the target size to $9 \times 9$, and the target pixels occupy 18.37% of the image. So in this case, that target is no longer a "small target". As claimed in previous section, the target is generated by the "randn" function with the mean value set to 10 and STD to 1 for all the 100 bands. Here, the degree of target spectral variation is changed by adjusting the STD value from 0 to 1 with an interval of 0.1. Again, we select the upper left corner pixel of the target region for **d**, and the result is shown in Figure 6. It can be seen that as the degree of target spectral variation increases, the detection accuracy gradually decreases. That is to say, the greater the spectral variation between target pixels is, the lower the detection accuracy is, and vice versa. An extreme example is that if there is no spectral variation between targets, even the target is not "small", all the targets can be extracted. For example, in Figure 6, when target's STD = 0, even the target contains 81 pixels, all pixels are extracted by CEM with AUC = 1. Therefore, we propose the following theorem:

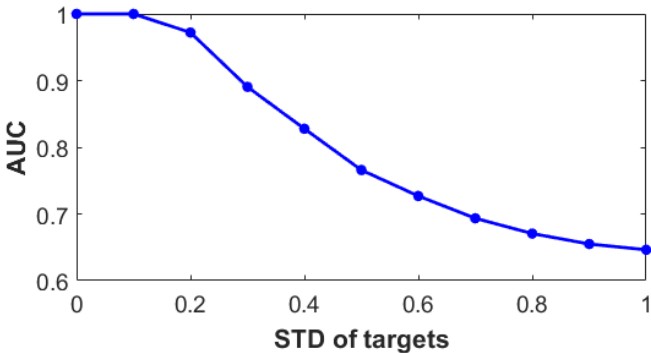

**Figure 6.** The target detection accuracy (AUC) vs. the standard deviation (STD) of the target pixels using the simulated data in Figure 5.

**Theorem 3.** *Assuming that the background pixels are fixed, when the spectra of the target pixels are the same, the target size has no effect on CEM's output.*

Please refer to Appendix A for the detailed proof of Theorem 3. Theorem 3 indicates that if all target pixels have the same spectrum, then the detection results will be the same regardless of the target size. Here we use a 2-D simulated data for illustration, which is generated in the same way as Figure 5. The distribution of target and background pixels is shown in Figure 7a, and the ground truth is shown in Figure 7b. The size of the data are $5 \times 6$, where the rightmost column represents the target, consisting of 5 pixels with identical spectra. To verify Theorem 3, we add a column of target pixels on the right side of the image, as shown in Figure 7c. The number of target pixels increases to 10, yet the targets' spectra remain all the same. The data matrices of the two simulated datasets are.

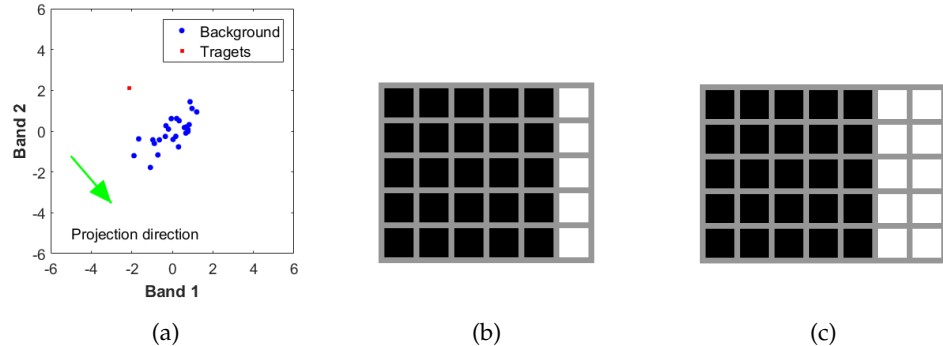

(a)                  (b)                  (c)

**Figure 7.** The distribution of the datasets to test the impact of target size to the result of target detection when the spectra of targets are all the same (**a**), the ground truth with $N_t = 5$ ((**b**), Background: black; Target: white), and the ground truth with $N_t = 10$ (**c**).

$$
\mathbf{X}_1 = \begin{bmatrix} 0.7679 & 0.0374 & 0.3361 & 0.7711 & -1.8904 & -0.7118 & -1.6585 & 0.6659 \\ -0.0232 & -0.4055 & 0.5126 & 0.0689 & -1.2011 & -1.1649 & -0.3794 & -0.0976 \end{bmatrix} \cdots
$$

$$
\begin{array}{cccccccc} 0.9692 & 0.3067 & 0.8302 & 0.5955 & \text{-}0.8855 & -0.0520 & -0.6379 & 1.2041 \\ 1.1100 & -0.7689 & 0.3186 & 0.1760 & \text{-}0.6017 & 0.6141 & -0.4254 & 0.9403 \end{array} \cdots
$$

$$
\begin{array}{cccccccc} -0.3113 & 0.8744 & -0.3431 & 0.2148 & -0.1903 & -1.0808 & -0.9630 & 0.6655 \\ 0.2653 & 1.4393 & -0.2581 & 0.6188 & 0.1015 & -1.7787 & -0.4259 & 0.2008 \end{array} \cdots
$$

$$
\begin{array}{ccccccc} 0.1747 & -2.1213 & -2.1213 & -2.1213 & -2.1213 & -2.1213 & -2.1213 \\ -0.2523 & 2.1213 & 2.1213 & 2.1213 & 2.1213 & 2.1213 & 2.1213 \end{array}
$$

which is a $2 \times 30$ matrix,

$$\mathbf{X}_2 = \begin{bmatrix} 0.7679 & 0.0374 & 0.3361 & 0.7711 & -1.8904 & -0.7118 & -1.6585 & 0.6659 \\ -0.0232 & -0.4055 & 0.5126 & 0.0689 & -1.2011 & -1.1649 & -0.3794 & -0.0976 \end{bmatrix} \cdots$$

$$\begin{matrix} 0.9692 & 0.3067 & 0.8302 & 0.5955 & -0.8855 & -0.0520 & -0.6379 & 1.2041 \\ 1.1100 & -0.7689 & 0.3186 & 0.1760 & -0.6017 & 0.6141 & -0.4254 & 0.9403 \end{matrix} \cdots$$

$$\begin{matrix} -0.3113 & 0.8744 & -0.3431 & 0.2148 & -0.1903 & -1.0808 & -0.9630 & 0.6655 \\ 0.2653 & 1.4393 & -0.2581 & 0.6188 & 0.1015 & -1.7787 & -0.4259 & 0.2008 \end{matrix} \cdots$$

$$\begin{matrix} 0.1747 & -2.1213 & -2.1213 & -2.1213 & -2.1213 & -2.1213 & -2.1213 & -2.1213 \\ -0.2523 & 2.1213 & 2.1213 & 2.1213 & 2.1213 & 2.1213 & 2.1213 & 2.1213 \end{matrix} \cdots$$

$$\begin{bmatrix} -2.1213 & -2.1213 & -2.1213 \\ 2.1213 & 2.1213 & 2.1213 \end{bmatrix}$$

which is a $2 \times 35$ matrix, and the targets' spectra are all the same,

$$\mathbf{d} = [-2.1213 \ 2.1213]^{\mathrm{T}}$$

Assuming the CEM detectors for the two datasets are $\mathbf{w}_1$ and $\mathbf{w}_2$, then according to (1), we can obtain $\mathbf{w}_1 = [-0.2187 \ 0.2527]^{\mathrm{T}}$ and $\mathbf{w}_2 = [-0.2187 \ 0.2527]^{\mathrm{T}}$. That is $\mathbf{w}_1 = \mathbf{w}_2$. This result shows that when the target spectra are identical, adding or removing target pixels has no effect on the detection result. (However, it should be noted that if the image size is fixed, increasing the target size will lead to a decrease in the number of background pixels. So even if the target spectra are all the same, adding more target pixels will cause a change in $\mathbf{R}$, and further cause a change in the CEM detector (see (A8) in Appendix A). Yet, as the image size increases, this change caused by the change of target size will decrease. When the image size tends to infinity, even without the premise that background pixels are fixed, Theorem 3 still holds.)

Therefore, under the circumstance that the target and background are linearly separable, if the target spectra are all the same, any target pixel can be selected as the representative one, regardless of the target size. That is to say, the size of the target is not the fundamental factor that affects the performance of target detection. The essential factor is the degree of variation in the target spectra. If the spectral similarity between targets is high, the influence of the target's size is relatively small. Conversely, if the target spectra differ greatly, the detection performance will be poor when the target size is large or when $N_t$ is large.

### 2.4. The Number of Bands

In addition to the three factors discussed above, the number of bands can also affect the result of target detection. For CEM, adding any band that is linearly independent of the original data can reduce the average output energy of the detector [75]. Suppose that $\Omega \subset (1, 2, \ldots, L)$ is an arbitrary subset of the band index set; $\mathbf{R}_\Omega$ and $\mathbf{d}_\Omega$ are the corresponding auto-correlation matrix and target spectral vector. We can have the following theorem:

**Theorem 4.** *The output energy from full bands is always less than that from the partial bands, i.e.,*

$$\frac{1}{\mathbf{d}^{\mathrm{T}} \mathbf{R}^{-1} \mathbf{d}} < \frac{1}{\mathbf{d}_\Omega^{\mathrm{T}} \mathbf{R}_\Omega^{-1} \mathbf{d}_\Omega}. \tag{6}$$

For a detailed proof of Theorem 4, please refer to reference [75]. Next, we will use a simulated data to illustrate this conclusion. The simulation data are generated in the same way as Figure 5, with a size of $21 \times 21$ (i.e., $N = 441$). The target size is set to $3 \times 3$, and the upper left pixel is selected as $\mathbf{d}$. Here, we gradually increase the number of bands from 20 to 441. From Figure 8a, one can see that the average filter output energy gradually decreases as the number of bands increases, verifying the conclusion of Theorem 4.

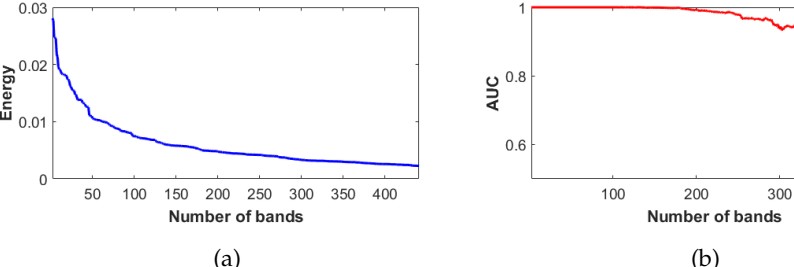

**Figure 8.** Impact of the number of bands to the result of CEM. (**a**) Average filter output energy vs. the number of bands. (**b**) AUC vs. the number of bands.

Therefore, adding bands can improve the performance of target detection from the perspective of average filter output energy. According Theorem 4, evening adding noisy bands can reduce the average filter output energy of the CEM detector, but noisy bands do not provide any useful information. It can be expected that adding noisy cannot improve the detection performance at all. Therefore, how to increase the number of bands or add bands or feature bands in practical applications is a crucial issue. It has be found that adding diagnostic bands that help to distinguish target pixels and background pixels to the original data cube can improve the target detection accuracy. For instance, incorporating water indices and water similarity bands into the original Landsat data (usually 6 or 7 bands) can effectively improve water extraction accuracy [76]. That is to say, only adding.

However, many scholars have found that reducing redundant bands can increase the separability of target pixels and background pixels [75], and the famous "Hughes phenomenon" also exists in hyperspectral target detection and classification [77]. Figure 9 shows the detection results corresponding to $L = 20, 200, 441$. When there are 20 bands, CEM can extract all target pixels. As the number of bands increases, the performance of the CEM detector does not improve, but instead becomes worse. For example, when there are 200 bands, except for the upper left corner target pixel, the output values of the other target pixels are difficult to be distinguished from those of background pixels. When there are 441 bands (i.e., $L = N$), only one target pixel can be extracted as all other target pixels (including other target points) are suppressed. Figure 8b shows the trend of target detection accuracy with the increase in the number of bands. Unlike the average filter output energy, the target detection accuracy exhibits an overall decrease trend when the number of bands is larger than 150. When $L = N$, the AUC value decreases to the lowest.

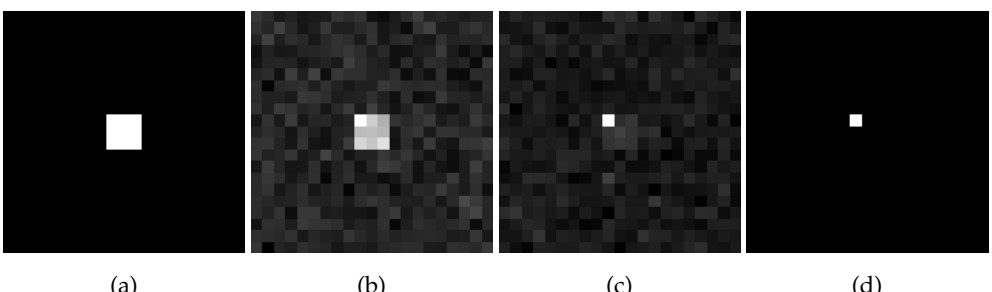

**Figure 9.** Results of CEM with different numbers of bands on the simulated data which is the same as Figure 8. (**a**) Ground truth. (**b**) $L = 20$. (**c**) $L = 200$. (**d**) $L = N(441)$.

That is to say, in the extreme case when $L = N$, only the target pixel which is selected as **d** can be extracted while all other pixels (including other target pixels) are completely suppressed as background. In this case, we have the following theorem:

**Theorem 5.** *When $L = N$ and the bands are linearly independent, only the output of the CEM detector on the target pixel selected as **d** is equal to 1, and the output values on the remaining pixels (including other target pixels) are all equal to 0.*

The proof of Theorem 5 can be found in Appendix B. Theorem 5 shows that if there exists spectral variation between target pixels, increasing the number of bands will increase the spectral difference between the selected target pixel (i.e., **d**) and other target pixels. As a result, the CEM detector will suppress other target pixels as well.

So, dimension reduction techniques, such as feature extraction and band selection, are often necessary. The former is to combine bands to new feature bands [69,78] while the latter is to select a subset of bands. Since band selection can retain the physical meaning of bands, they are commonly used for target detection. There are two main categories of band selection methods: supervised and unsupervised. The former rely on criteria like class separability and require training samples. The latter do not require prior knowledge and can be divided into ranking-based and clustering-based approaches. Ranking-based methods use a criterion to measure band importance [79–81], while clustering-based methods aim to select bands that are close to cluster centers [82,83]. In many cases, dimensional reduction can further improve the detection performance [69,84].

Therefore, the number of bands does not have a direct relationship with the target detection performance. In the process of hyperspectral target detection, whether we need to increase or decrease the number of bands depends on the data distribution, i.e., the degree of spectral variation in the target and the separability between the target and the background.

Next, we will discuss how to use the aforementioned conclusions and knowledge to guide real hyperspectral target detection with varying levels of difficulty.

## 3. Strategies to Improve the Target Detection Accuracy under Different Conditions

In the previous section, we have conducted a theoretical analysis on how the four aspects (i.e., the data origin, target size, target spectral variation, and the number of bands) affect the performance of target detection. For real hyperspectral target detection, based on the degree of target spectral variation and the separability between the target and the background, it can be generally divided into four cases:

(1) The target spectral variation is small while the spectral difference between target and background is large;
(2) The target spectral variation is small while the spectral difference between target and background is small;
(3) The target spectral variation is large while the spectral difference between target and background is large;
(4) The target spectral variation is large while the spectral difference between target and background is small.

In this section, we use real hyperspectral datasets to conduct a detailed analysis of the four cases listed above and provide corresponding strategies based on the following aspects,

(1) Data origin: Since the issue of the data origin has already been solved theoretically (see Section 2.1), we suggest to centralize the data firstly (i.e., subtract the mean vector from the data firstly). Therefore, for the following experiments, all the data are centralized firstly.
(2) Target size and target spectral variation: If the single-target detection method is chosen, the problem becomes how to select a proper spectrum to represent the target pixels. If the multiple-target detection method is chosen, the problem turns to how to choose a proper spectrum for each category of the target.
(3) Number of bands: We mainly considers how to select proper bands and/or increase feature bands.

As the data origin issue has already been resolved in the previous section, the discussion in this section mainly focuses on the latter two aspects.

*3.1. The Target Spectral Variation Is Small While the Spectral Difference between Target and Background Is Large*

In this part, we mainly discuss how to deal with the issue when there is small spectral variation between target pixels and a significant spectral difference between target and background spectra. Here, a real hyperspectral image of $200 \times 200$ pixels, as shown in Figure 10a is used for illustration. It was collected by the Operational Modular Imaging Spectrometer-II, which is a hyperspectral imaging system developed by Shanghai Institute of Technical Physics, Chinese Academy of Sciences. The data were acquired in the city of Xi'an, China in 2003, and are composed of 64 bands from 460 nm to 10,250 nm with a spatial resolution of 3.6 m. A small human-made object, marked by an arrow in the scene is selected as the target of interest, and the manually determined ground truth map based on the field survey is shown in Figure 10b.

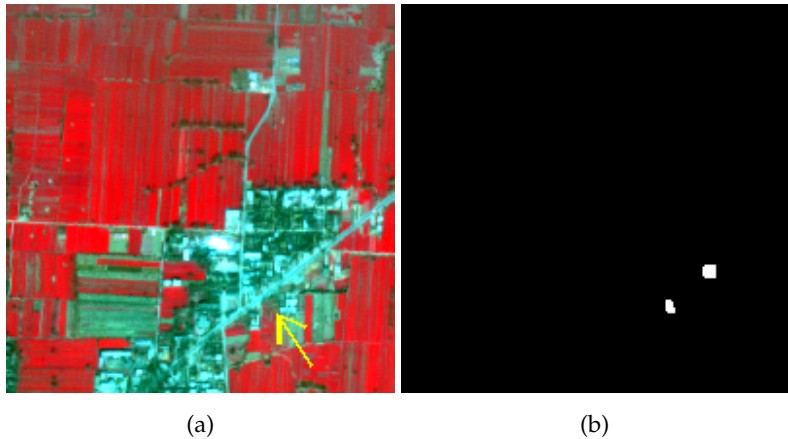

　　　　　　　(a)　　　　　　　　　　　　　　　　　(b)

**Figure 10.** The false color image of the Xi'an data ((**a**), R: 857.9 nm, G: 669.6 nm, B: 559.6 nm) and the ground truth (**b**).

Firstly, we will discuss the impact of the selection of the representative target spectrum (i.e., **d**) on the final target detection result. We select nine target pixels as the representative spectrum, and the corresponding spectral curves are shown in Figure 11a. It can be seen that the nine target pixels have very similar spectra, which satisfies the constraint of this case, i.e., "The target spectral variation is small". Then nine CEM detectors are constructed using each of these nine spectra individually as **d**, and the corresponding AUC values are shown in Figure 11b. Some detection results are shown in Figure 11c–e. It can be seen that the detection accuracy values corresponding to the nine target pixels are all very high, indicating that the choice of **d** has little effect on the detection results. That is to say, the requirement for the choice of the representative spectrum is relatively low for this case, and any target spectrum selected as the representative spectrum can achieve a satisfactory detection result.

Next, we will study the impact of band selection on the detection results. Selecting Target 5 as **d** (Figure 12a), we extract the first 5, 10, 15, . . . , 60 bands for target detection. And the corresponding accuracy values are plotted in Figure 12b (the last one being the detection accuracy using all bands (i.e., $L = 64$)). It can be seen that when $L \geq 15$, the accuracy of target detection can remain stable at a high level, indicating that the background can be well suppressed when more than 15 bands are used (Figure 12c–e). This experimental result shows that under this condition, there is no high requirement for the band selection. However, if there are too few bands, such as when $L \leq 10$, the target detection accuracy will significantly decrease.

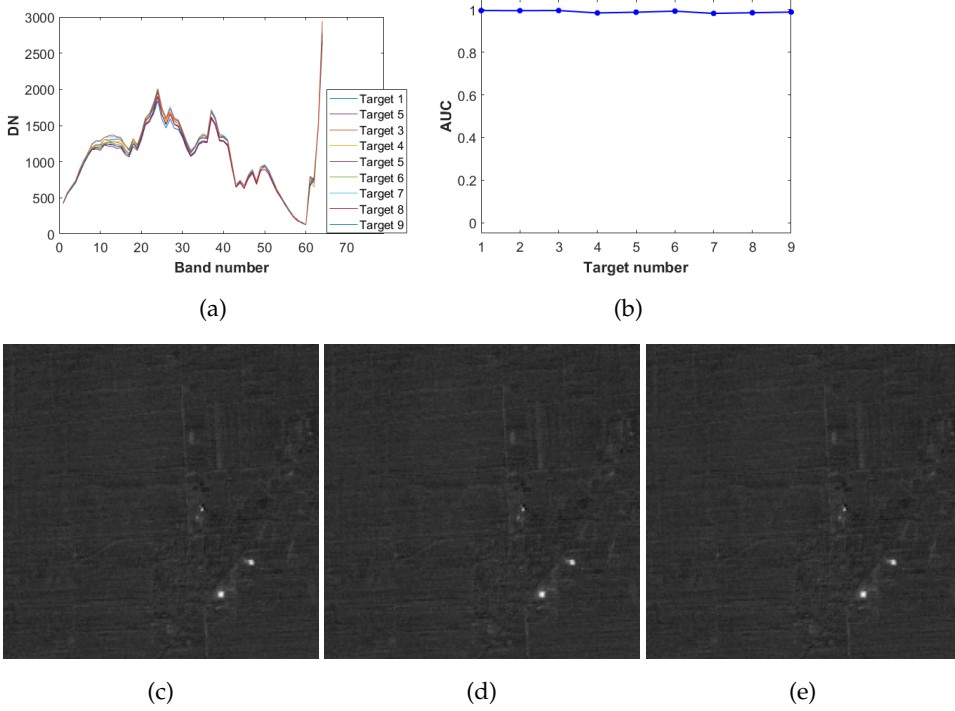

(c)   (d)   (e)

**Figure 11.** Target detection results for Xi'an data when using different target pixels as **d**. (**a**) The spectra of 9 selected target pixels. (**b**) The AUC values corresponding to the 9 target pixels. (**c**) Output of CEM when choosing Target 3 as **d**. (**d**) Output of CEM when choosing Target 6 as **d**. (**e**) Output of CEM when choosing Target 9 as **d**.

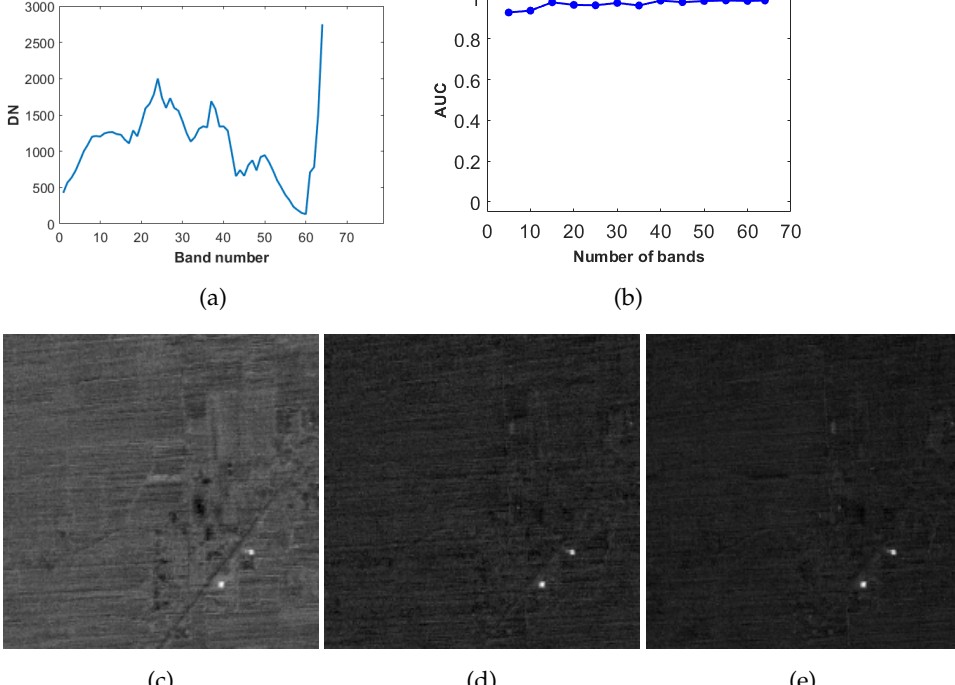

(c)   (d)   (e)

**Figure 12.** Target detection results for Xi'an data with different numbers of bands. (**a**) The spectrum of Target 5. (**b**) The AUC values vs. the number of bands ($L$). (**c**) $L = 15$. (**d**) $L = 35$. (**e**) $L = 64$.

In summary, for this case that "The target spectral variation is small while the spectral difference between target and background is large", the requirements for selecting representative spectrum and band selection are both low. It is recommended that:

(1)     Selection of **d**: Select the central pixel of the target region as the representative one, avoiding to select the spectral mixing pixels at the boundary of the target region as **d**; or select the mean spectrum of all target pixels as **d**.

(2)     Band operation: If there is no requirement on computational efficiency, all the bands except for the noise or bad bands should be selected; otherwise, band selection methods can be used before the target detection to reduce the number of bands. Furthermore, if a higher target detection accuracy is required, one can add more feature bands which are helpful to differentiate the target from the background.

### 3.2. The Target Spectral Variation Is Small While the Spectral Difference between Target and Background Is Small

Next, we will discuss the second case, i.e., "The target spectral variation is small while the spectral difference between target and background is small". According to previous discussions, we can see that if the spectral variation of target pixels is small, the choice of **d** has a minor impact on the detection result. Therefore, we only focus on the influence of band selection in this case. Here, the commonly used Salinas data are utilized (data download address: www.ehu.es/ccwintco/index.php/Hyperspectral_ Remote_Sensing_Scenes (accessed on 30 July 2023)). These data were captured by the Airborne Visible Infrared Imaging Spectrometer (AVIRIS) hyperspectral sensor in Salinas Valley, California with a size of $512 \times 217$ and contains 224 bands ranging from 370 to 2510 nm. The dataset includes 15 classes, as shown in Figure 13. This study focuses on Class 5 ("Fallow_smooth"), which has a good internal spectral consistency and thereby meets the requirement of "The target spectral variation is small". Additionally, other bare soil fields exist in this dataset besides Class 5: Class 3 ("Fallow"), Class 4 ("Fallow_rough_plow"), and Class 9 ("Soil_vinyard_develop"). Thus, it also meets the requirement of "the spectral differences between target and background is small". In our experiment, we take the average spectrum of all Class 5 pixels as **d**.

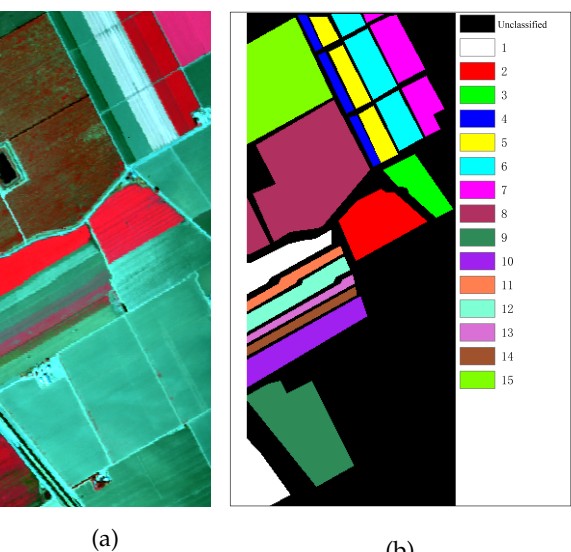

(a)                                                        (b)

**Figure 13.** The false color image of the Salinas data ((**a**), R: Band 55, G: Band 33, B: Band 19) and the ground truth (**b**).

To investigate the impact of the number of bands on target detection, we select the first 10, 20, ..., 220 bands, as well as all bands ($L = 224$), from the original data for target detection. The results are shown in Figure 14 (where $L = 40, 70, 224$, respectively, correspond to visible, visible to near-infrared, and all bands). It can be seen that the accuracy of target detection increases as the number of bands. Particularly, when $L < 40$, the increase of AUC as the number of bands is significant. Comparing Figure 14c–e, it can be found that adding short-wave infrared bands can effectively distinguish Class 5 from

other bare soil classes. It is worth noting that although, when $L \geq 40$, the increase rate in target detection accuracy vs. $L$ becomes slower, the upward trend does not stop even when $L$ is large. This indicates that if new bands that are useful for distinguishing target from background are added, the detection accuracy of Class 5 can be further improved.

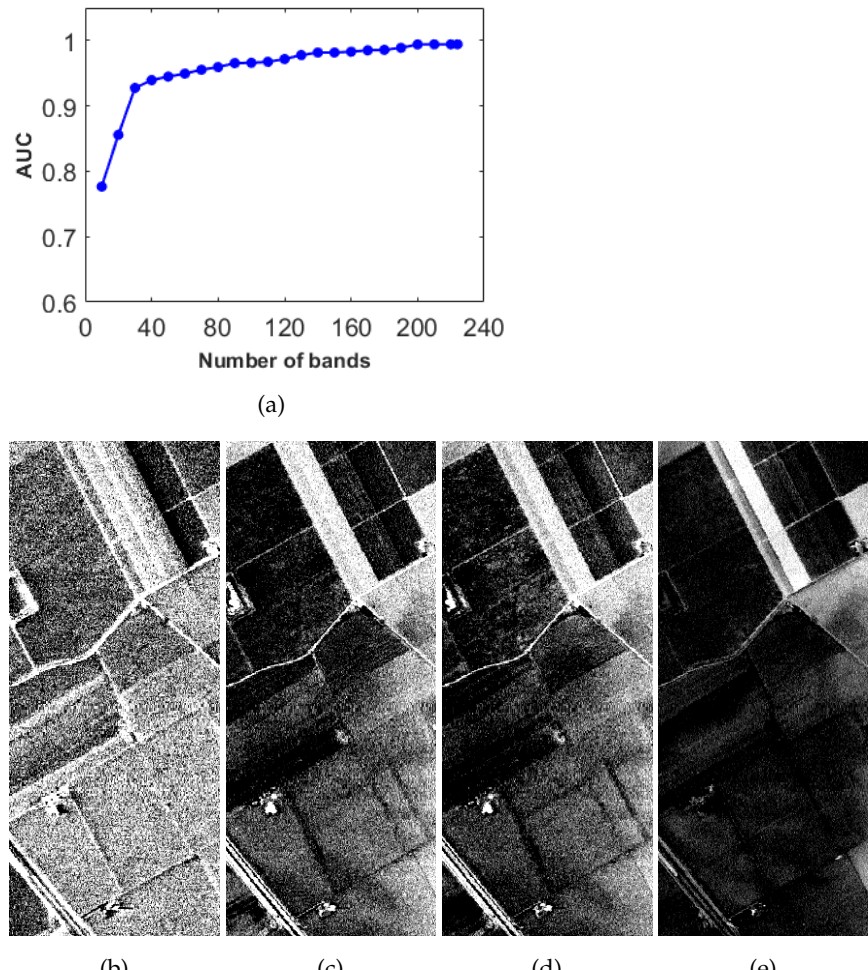

(a)

(b)        (c)        (d)        (e)

**Figure 14.** Effect of the number of bands to the target detection result on the Salinas data. (**a**) AUC vs. the number of bands. (**b**) $L = 10$. (**c**) $L = 40$. (**d**) $L = 70$. (**e**) $L = 224$.

Therefore, for the case "The target spectral variation is small while the spectral difference between target and background is small", the following recommendations are suggested:

(1) Selection of **d**: Similar to Case 1, try to avoid selecting spectral mixing pixels at the target boundary as **d**. Instead, pixels from the central region of the target or the mean spectrum can be chosen as **d**.

(2) Band operation: By adding useful bands, the difference between the target and background can be increased, so as to improve the accuracy of the target detection.

*3.3. The Target Spectral Variation Is Large While the Spectral Difference between Target and Background Is Large*

Further, in this section we will discuss the case that the target spectral variation is large while the spectral difference between the target and background is large. According to Theorem 5, when there are too many bands, the target detector is prone to mis-classify the "brother pixels" of **d**. That is to say, other target pixels other than the one selected as **d** will be suppressed as background too. Therefore, if there exist a big spectral variation between target pixels' spectra, there is a high probability that the detector will mis-classify

the "brother pixels" no matter which target pixel is chosen as **d** when the data have many bands. In other words, it is difficult to extract all target pixels in this case. Therefore, in the following we will focus on two aspects: (1) band operation and (2) the selection of **d**.

For illustration, we select a Gaofen (GF)-5 hyperspectral data as an example, as shown in Figure 15a (which is downloaded at https://www.cheosgrid.org.cn/ (accessed on 30 July 2023)). The data were obtained by the visible to short-wave infrared hyperspectral camera on the GF-5 satellite on 17 April 2019, and contains 330 bands. In this study, only the 150 visible near-infrared bands with a spectral resolution of 5 nm and a size of $400 \times 400$ are used. The scene is located in Xinjiang, China, and the main land covers are bare soil, water, and vegetation. In this experiment, we take the "vegetation" as the target. Generally in the false-color image, vegetation pixels appear in a red color. From Figure 15a, it can be seen that the color of vegetation varies greatly in this area, including light red, bright red, dark red, and gray red, which correspond to different types and/or abundances of vegetation. Therefore, this satisfies the case discussed in this section—"The target spectral variation is large". The ground truth of vegetation is obtained by using a threshold method on the NDVI band. That is, any pixel with $NDVI \geq 0.2$ is marked as vegetation, as shown in Figure 15b.

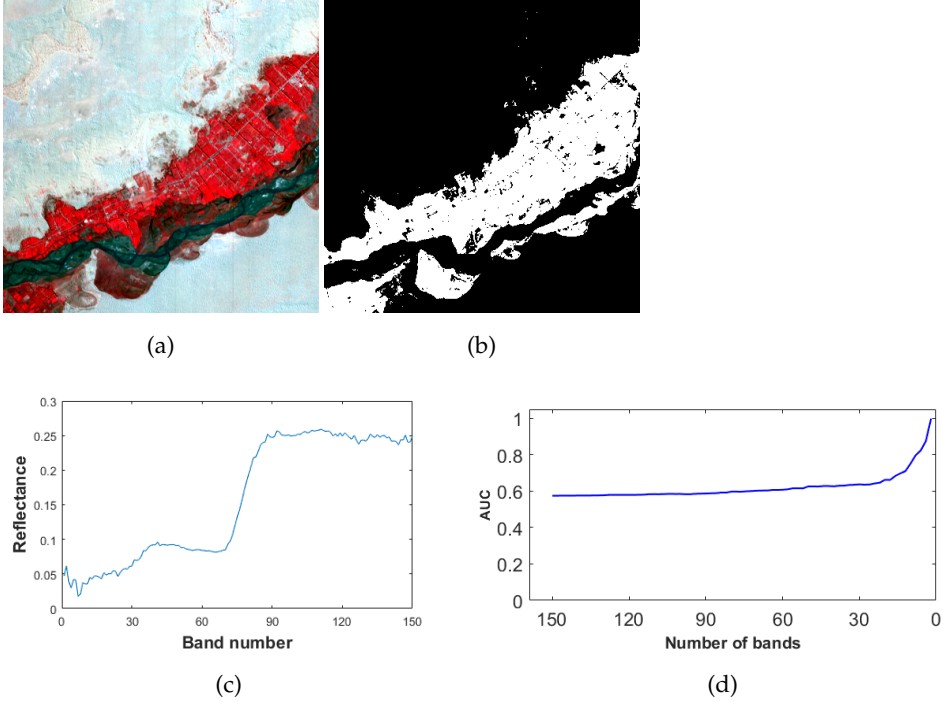

(a)    (b)

(c)    (d)

**Figure 15.** Impact of the number of bands to the result of target detection for the Xinjiang data. (**a**) The false color image (R: NIR band, G: red band, B: green band). (**b**) Ground truth. (**c**) The spectral curve of the selected target pixel. (**d**) AUC vs. the number of bands.

We randomly choose a vegetation pixel as **d** (Figure 15c) and conduct the target detection on the full band data, as shown in Figure 16a. It can be seen that almost no vegetation pixels can be detected, which again verifies the conclusion of Theorem 5. Furthermore, we use the a band selection method, named Fast Volume-Gradient-Based Band Selection (FastVGBS) [81] for band selection. We gradually reduce the number of bands from 150 to 2 with a step of 2. The corresponding detection accuracy is shown in Figure 15d. The target detection results with $L = 50, 20, 10, 6, 2$ are shown in Figure 16b, Figure 16c, Figure 16d, Figure 16e, Figure 16f, respectively, and the corresponding selected band indices are tabulated in Table 2. From Figure 15d, it can be seen that as the number of bands decreases, the accuracy of target detection slowly increases at first, however, when $L \leq 20$, the AUC rapidly increases. Especially when the number is reduced to 2, the accuracy reaches the maximum, and the corresponding target detection result is shown in Figure 16f. One

can observe that most vegetation pixels are extracted. It means that we only need two diagnostic spectral bands to distinguish vegetation from other ground objects. Similarly, for other ground objects having large spectral differences from the background, only a few diagnostic bands are needed for target detection. Involving too many bands will instead hamper to extract all target pixels.

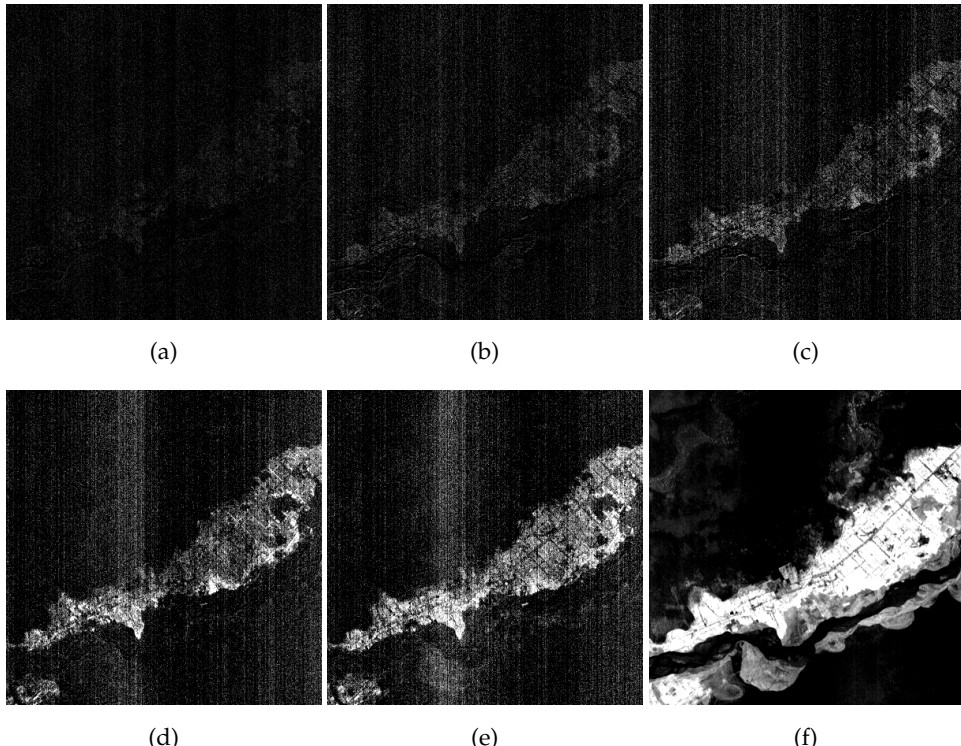

**Figure 16.** Target detection results with different numbers of bands for the Xinjiang data. (**a**) 150 bands. (**b**) 50 bands. (**c**) 20 bands. (**d**) 10 bands. (**e**) 6 bands. (**f**) 2 bands.

**Table 2.** Band indices selected by FastVGBS for the Xinjiang data with $L = 50$, 20, 10, 6 and 2. Note that the visible range: Band 1–85, and the near infrared range: Band 86–150.

| $L$ | Band Indices |
|---|---|
| 50 | 1 2 3 4 5 6 7 8 9 10 11 34 68 78 87 88 101 102 119 120 121 122 123 124 125 126 127 128 129 130 131 132 133 134 135 136 137 138 139 140 141 142 143 144 145 146 147 148 149 150 |
| 20 | 1 2 3 4 7 68 78 88 127 128 129 130 133 143 145 146 147 148 149 150 |
| 10 | 1 2 3 68 78 88 128 146 149 150 |
| 6 | 1 2 68 88 146 150 |
| 2 | 68 150 |

Next, we will discuss the selection of **d**. Since the spectral heterogeneity between target pixels is large, it is difficult to select any target spectrum to represent the spectral characteristics of all target pixels. Therefore, we can select more than one target pixel as **d**, and the multi-target detection can be used instead of the single-target detection. To extract all types of vegetation targets, we classify the target pixels into six categories according to their NDVI values:

- Target 1: NDVI $\in [0.2, 0.3)$;
- Target 2: NDVI $\in [0.3, 0.4)$;
- Target 3: NDVI $\in [0.4, 0.5)$;
- Target 4: NDVI $\in [0.5, 0.6)$;
- Target 5: NDVI $\in [0.6, 0.7)$;

- Target 6: NDVI $\in [0.7, 1]$.

Then, the mean spectrum of each category is calculated, as shown in Figure 17a. Three multi-target detection algorithms are used for vegetation detection, the Sum CEM [41], Winner-Take-All CEM (WTACEM) [41], and Multiple Target CEM (MTCEM) [41]. The results are shown in Figure 17 and the accuracy evaluation results are tabulated in Table 3. It can be seen that compared with the single-target detection algorithm CEM (Figure 16a), all the three multi-target detection algorithms can extract all vegetation pixels very well, and thus the extraction accuracy is much higher than that of the single-target detection method.

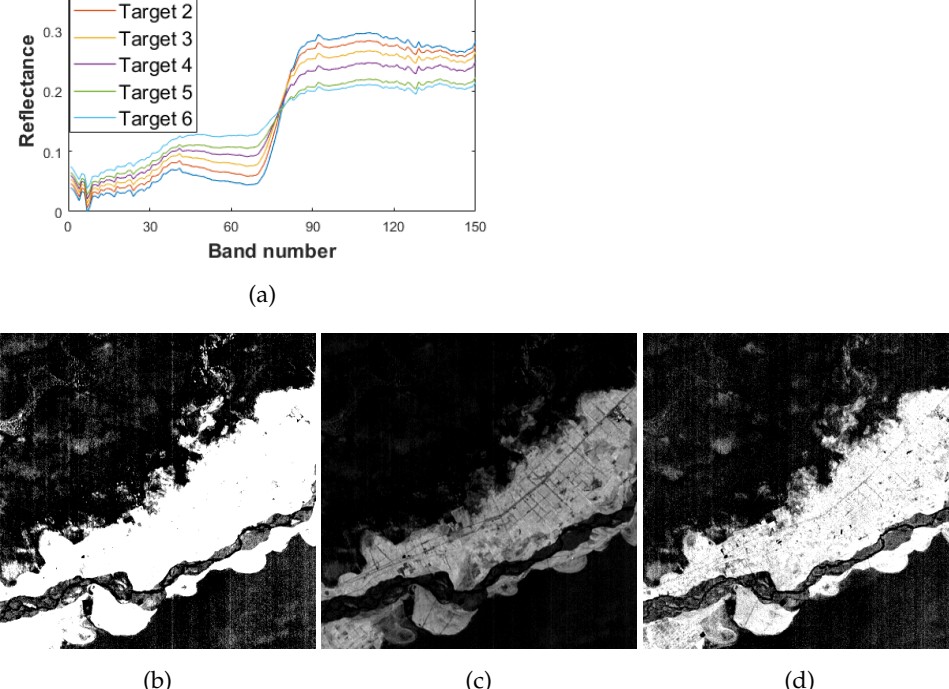

(a)

      (b)               (c)               (d)

**Figure 17.** Target detection results using the multi-target detection methods for the Xinjiang data. (**a**) The spectra of 6 target pixels. (**b**) SCEM. (**c**) WTACEM. (**d**) MTCEM.

**Table 3.** AUC values of the single-target detection method (CEM) and multi-target detection methods (SCEM, WTACEM, and MTCEM) for the Xinjiang data.

| Algorithm | CEM | SCEM | WTACEM | MTCEM |
|:---:|:---:|:---:|:---:|:---:|
| AUC | 0.5749 | 0.9988 | 0.9988 | 0.9841 |

In summary, for the case of "The target spectral variation is large while the spectral difference between target and background is large", the following recommendations are suggested:

(1) If using a single-target detection algorithm, the number of bands can be reduced by using the band selection method to remove redundant bands. Since the target pixels' spectra vary greatly, it is recommended to use the mean value of all target pixels as **d**.

(2) Multi-target detection algorithms can be used instead of the single-target detection algorithms, but it should be noted that the multiple representative spectra of target should cover all target categories. To improve computational efficiency, redundant bands can also be removed when using multi-target detection algorithms.

*3.4. The Target Spectral Variation Is Large While the Spectral Difference between Target and Background Is Small*

Finally, we will discuss the fourth case that "The target spectral variation is large while the spectral difference between target and background is small". In this case, the difficulty to extract all target pixels while suppressing all background pixels is the largest. Based on previous discussions, if "the target spectral variation is large", we should reduce the number of bands and add useful bands; yet if "the spectral difference between target and background is small", we should increase the number of bands. Therefore, combining these two situations, we propose the following suggestions:

(1) When using the single target detection method, the detection accuracy can be improved mainly by band operation. On one hand, we should reduce redundant bands and on the other hand, we should add useful feature bands that can increase the difference between target and background. The average spectrum should be selected for **d**.

(2) When using the multi-target detection method, the representative spectra should cover all types of target pixels. Moreover, it is also recommended to simultaneously conduct band selection before performing multi-target detection, so as to reduce redundant bands and increase useful feature bands, which can further improve the detection accuracy.

In addition, the spectral variation can also be decreased by increasing spatial continuity. Here, the commonly used Indian Pines data are used for illustration. The data were collected by the AVIRIS hyperspectral sensor on 12 June 1992, with a size of $145 \times 145$ and containing 220 bands from the visible to shortwave-infrared range. The Indian Pines image contains 16 classes, as shown in Figure 18. It can be seen that there is high spectral similarity between many classes. For example, Class 2 (corn-notill), 3 (corn-mintill), 10 (soybean-notill), and 11 (soybean-mintill) all exhibit the spectral characteristics of bare soil; Class 5 (grass-pasture), 6 (grass-trees), 13 (wheat), and 14 (woods) all exhibit the spectral characteristics of vegetation. Therefore, detecting any of these categories will lead to the problem of "The spectral difference between target and background is small". On the other hand, there is a low consistency within many classes, such as Class 11 (soybean-mintill) and Class 14 (woods) (see Figure 18).

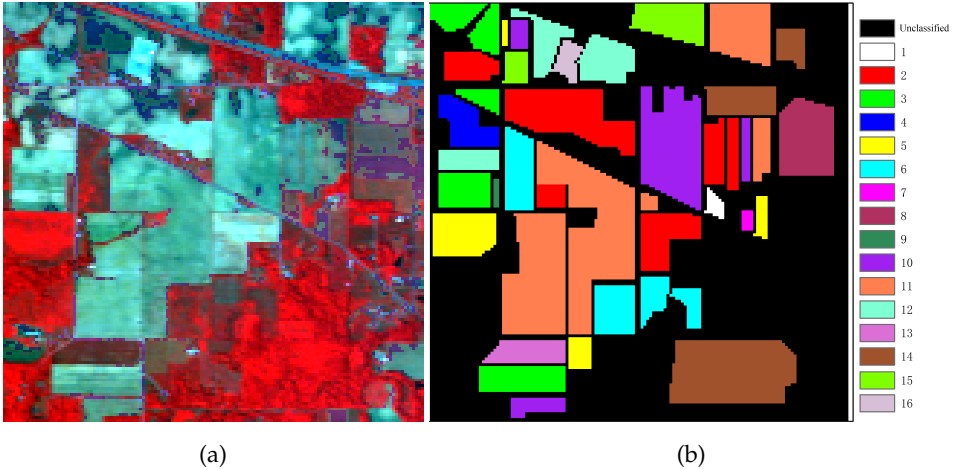

(a) (b)

**Figure 18.** The false color image ((**a**), R: Band 42, G: Band 30, B: Band 8) and the ground truth (**b**) of the Indian Pines data.

Next, we first use the Normalized MF Weight (NMFW) method [85] to remove 40 useless bands (noise or bad bands), and then use the spectral recognition spatial smooth hyperspectral filter [22] to increase the spatial consistency between land covers. Finally, the target detection is performed for each category, and the corresponding accuracy metrics are

shown in Table 4. It can be seen that for all classes, performing spatial filtering can improve the detection accuracy.

**Table 4.** Comparisons between AUC metrics using original data and filtered data for the Indian Pines data.

| Class No. | 1 | 2 | 3 | 4 | 5 | 6 | 7 | 8 |
|---|---|---|---|---|---|---|---|---|
| Original | 0.9985 | 0.9589 | 0.9191 | 0.9676 | 0.9487 | 0.9828 | 0.9992 | 0.9966 |
| Filtered | 0.9991 | 0.9706 | 0.9334 | 0.9738 | 0.9656 | 0.9883 | 0.9996 | 0.9976 |
| Class No. | 9 | 10 | 11 | 12 | 13 | 14 | 15 | 16 |
| Original | 0.9994 | 0.9448 | 0.9417 | 0.9784 | 0.9992 | 0.9619 | 0.9667 | 0.9982 |
| Filtered | 0.9996 | 0.9613 | 0.9523 | 0.9829 | 0.9996 | 0.9689 | 0.9792 | 0.9990 |

Therefore, for the case that "The target spectral variation is large while the spectral difference between target and background is small", it is also recommended to improve the spatial consistency of the target and background by using spatial filtering algorithms.

## 4. Conclusions

Target detection is an important research field in hyperspectral remote sensing. For statistic-based target detection methods, we theoretically discuss the impact of data origin, target size, target spectral variation, and the number of bands on the performance of the target detection. The main conclusions are as follows:

(1) The data origin will affect the performance of target detection. However, there exist optimal data origins in terms of average filter output energy, named CE points. Yet the corresponding CE detector is equivalent to the MF detector, it is recommended to subtract the mean vector before conducting target detection.

(2) As the target size increases, the detection of the target may deteriorate because the influence of target pixels on the auto-correlation matrix becomes more significant. However, Theorem 3 states that the target detection results will remain unaffected under the premise that the target pixels has the same spectrum.

(3) The degree of target spectral variation is what actually affects the target detection result.

(4) From the perspective of the average filter output energy, adding more bands can achieve a better target detection. However, from the perspective of detection accuracy, too many bands may not necessarily be beneficial for target detection. In particular, when the spectral variation of the target is large, the disadvantage of having more bands becomes more apparent. This is because with more bands, detectors are more likely to mislabel other target pixels as background.

Furthermore, we conduct four series of experiments using real hyperspectral data and provides strategies on the selection of data origin and target spectrum, band operation, and spatial filtering for four common scenarios, as shown in Table 5.

Finally, researchers interested in hyperspectral target detection based on statistical information can download the codes of most algorithms used in this paper from the following GitHub repository: https://github.com/jiluyan/hyperspectral-target-detection.git (accessed on 30 July 2023).

**Table 5.** Four cases in target detection and the corresponding strategies.

| Case No. | Description (Difficulty Level) | Diagram | Strategies | | | |
|---|---|---|---|---|---|---|
| | | | Data Origin | Selection of Target Spectrum | Band Operation | Spatial Filter |
| 1 | The target spectral variation is small while the spectral difference between target and background is large (Easy) |  | Move the data to the mean vector of the data | Select the representative target spectrum or the mean vector of the target as **d**. | Reduce redundant or bad bands to increase the computational efficiency. | |
| 2 | The target spectral variation is small while the spectral difference between target and background is small (Normal) |  | Move the data to the mean vector of the data. | Select the representative target spectrum or the mean vector of the target as **d**. | Add useful bands. | |
| 3 | The target spectral variation is large while the spectral difference between target and background is large (A bit challenging) |  | Move the data to the mean vector of the data. | Select the representative target spectrum or the mean vector of the target as **d**. | Reduce redundant or bad bands to increase the computational efficiency. | |
| 4 | The target spectral variation is large while the spectral difference between target and background is small (Hard) |  | Move the data to the mean vector of the data. | Select the representative target spectrum or the mean vector of the target as **d**. | Reduce redundant or bad bands, and add useful bands. | Use the spatial filter to reduce the target spectral variation. |

**Author Contributions:** Experiment and writing: L.J.; methodology: X.G. All authors have read and agreed to the published version of the manuscript.

**Funding:** This research was funded by the National Key Research and Development Program of China (grant number 31400), and the Second Tibetan Plateau Scientific Expedition and Research Program (grant number 2019QZKK0206).

**Institutional Review Board Statement:** Not applicable.

**Informed Consent Statement:** Not applicable.

**Data Availability Statement:** Not applicable.

**Acknowledgments:** We would like to thank Shihang Chen from University of Chinese Academy of Sciences, China, Qinyu Zhao from the Hebei University, China for the help on the language and paper searching.

**Conflicts of Interest:** The authors declare no conflict of interest.

## Appendix A. The Proof of Theorem 3

**Proof.** Suppose $\mathbf{x}_i$ is the $i$th pixel of the image ($i = 1, 2, \ldots, N$, $N$ is the total number of pixels), $\mathbf{x}_{bj}$ is the $j$th background pixel ($j = 1, 2, \ldots, N_b$, $N_b$ is the total number of background pixels), and $\mathbf{d}_{tk}$ is the $k$th target pixel ($k = 1, 2, \ldots, N_t$, $N_t$ is the total number of target pixels). So, $N = N_b + N_t$. Since the spectra of the target pixels are the same, we can have $\mathbf{d}_1 = \mathbf{d}_2 = \cdots = \mathbf{d}_{N_t} = \mathbf{d}$. In order to calculate the detector of CEM, we firstly calculate the auto-correlation matrix $\mathbf{R}$:

$$\mathbf{R} = \frac{1}{N} \sum_{i=1}^{N} \mathbf{x}_i \mathbf{x}_i^{\mathrm{T}} = \frac{1}{N} \left( \sum_{j=1}^{N_b} \mathbf{x}_{bj} \mathbf{x}_{bj}^{\mathrm{T}} + \sum_{k=1}^{N_t} \mathbf{d}_k \mathbf{d}_k^{\mathrm{T}} \right) \tag{A1}$$

Let $\mathbf{R}_b = \frac{1}{N_b} \sum_{j=1}^{N_b} \mathbf{x}_{bj} \mathbf{x}_{bj}^{T}$, and substitute it into (A1), we can have:

$$\mathbf{R} = \frac{N_b}{N} \mathbf{R}_b + \frac{N_t}{N} \mathbf{d} \mathbf{d}^{\mathrm{T}} \tag{A2}$$

Then, let $\alpha = \frac{N_b}{N}$, $\beta = \frac{N_t}{N}$, and substitute them into the above equation, we can have a more concise expression of $\mathbf{R}$:

$$\mathbf{R} = \alpha \mathbf{R}_b + \beta \mathbf{d} \mathbf{d}^{\mathrm{T}} \tag{A3}$$

According to Sherman and Morrison Equation [86], the inversion of $\mathbf{R}$ can be computed as

$$\mathbf{R}^{-1} = \left( \alpha \mathbf{R}_b + \beta \mathbf{d} \mathbf{d}^{\mathrm{T}} \right)^{-1} = \frac{1}{\alpha} \mathbf{R}_b^{-1} - \frac{\frac{\beta}{\alpha^2} \mathbf{R}_b^{-1} \mathbf{d} \mathbf{d}^{\mathrm{T}} \mathbf{R}_b^{-1}}{1 + \frac{\beta}{\alpha} \mathbf{d}^{\mathrm{T}} \mathbf{R}_b^{-1} \mathbf{d}} \tag{A4}$$

Let $\gamma = \mathbf{d}^{\mathrm{T}} \mathbf{R}_b^{-1} \mathbf{d}$ and define the coefficient of second part of (A4) as $\delta$, which can be expressed by $\alpha, \beta$, and, $\gamma$ as:

$$\delta = -\frac{\beta}{\alpha^2 \left( 1 + \frac{\beta}{\alpha} \mathbf{d}^{\mathrm{T}} \mathbf{R}_b^{-1} \mathbf{d} \right)} = -\frac{\beta}{\alpha^2 (1 + \frac{\beta \gamma}{\alpha})} \tag{A5}$$

Then, $\mathbf{R}^{-1}$ can be expressed as:

$$\mathbf{R}^{-1} = \frac{1}{\alpha} \mathbf{R}_b^{-1} + \delta \mathbf{R}_b^{-1} \mathbf{d} \mathbf{d}^{\mathrm{T}} \mathbf{R}_b^{-1} \tag{A6}$$

According to (A6), the denominator of the CEM detector can be calculated as:

$$\mathbf{d}^{\mathrm{T}} \mathbf{R}^{-1} \mathbf{d} = \mathbf{d}^{\mathrm{T}} \left( \frac{1}{\alpha} \mathbf{R}_b^{-1} + \delta \mathbf{R}_b^{-1} \mathbf{d} \mathbf{d}^{\mathrm{T}} \mathbf{R}_b^{-1} \right) \mathbf{d} = \frac{1}{\alpha} \mathbf{d}^{\mathrm{T}} \mathbf{R}_b^{-1} \mathbf{d} + \delta \mathbf{d}^{\mathrm{T}} \mathbf{R}_b^{-1} \mathbf{d} \mathbf{d}^{\mathrm{T}} \mathbf{R}_b^{-1} \mathbf{d} = \frac{\gamma}{\alpha} + \delta \gamma^2 \tag{A7}$$

Then, the CEM detector in (1) can be recalculated as:

$$\begin{aligned} \mathbf{w}_{\mathrm{CEM}} &= \frac{\mathbf{R}^{-1} \mathbf{d}}{\mathbf{d}^{\mathrm{T}} \mathbf{R}^{-1} \mathbf{d}} = \frac{1}{\frac{\gamma}{\alpha} + \delta \gamma^2} \left( \frac{1}{\alpha} \mathbf{R}_b^{-1} + \delta \mathbf{R}_b^{-1} \mathbf{d} \mathbf{d}^{\mathrm{T}} \mathbf{R}_b^{-1} \right) \mathbf{d} \\ &= \frac{1}{\frac{\gamma}{\alpha} + \delta \gamma^2} \left( \frac{1}{\alpha} \mathbf{R}_b^{-1} \mathbf{d} + \delta \mathbf{R}_b^{-1} \mathbf{d} \mathbf{d}^{\mathrm{T}} \mathbf{R}_b^{-1} \mathbf{d} \right) \\ &= \frac{1}{\frac{\gamma}{\alpha} + \delta \gamma^2} \left( \frac{1}{\alpha} + \delta \gamma \right) \mathbf{R}_b^{-1} \mathbf{d} = \frac{1}{\gamma} \mathbf{R}_b^{-1} \mathbf{d} = \frac{\mathbf{R}_b^{-1} \mathbf{d}}{\mathbf{d}^{\mathrm{T}} \mathbf{R}_b^{-1} \mathbf{d}} \end{aligned} \tag{A8}$$

From the above expression, we can see that the direction of $\mathbf{w}_{\text{CEM}}$ is determined by $\mathbf{R}_{\text{b}}$ and $\mathbf{d}$. Therefore, when the spectra of the background are fixed and all the spectra of target pixels are the same, the direction of the CEM detector will not change. That is to say, when we add more target pixels with the same spectrum to the image, the target detection result will not change. $\square$

**Appendix B. Proof of Theorem 5**

**Proof.** Without loss of generality, we select the last target pixel $\mathbf{d}_{N_t}$ as the representative target pixel. For simplicity, let $\mathbf{d}_{N_t} = \mathbf{d}$. So the data matrix can be written as $\mathbf{X} = \begin{bmatrix} \mathbf{x}_1 \ \mathbf{x}_2 \ \dots \ \mathbf{x}_{N_b} \ \mathbf{d}_1 \ \mathbf{d}_2 \ \dots \ \mathbf{d}_{N_t-1} \ \mathbf{d} \end{bmatrix}$, and the corresponding output values of CEM detector as $\mathbf{y} = \begin{bmatrix} y_1^b \ y_2^b \ \dots \ y_{N_b}^b \ y_1^t \ y_2^t \ \dots \ y_{N_t-1}^t \ y_{N_t}^t \end{bmatrix}$. Firstly, the auto-correlation matrix of the data can be calculated by:

$$\mathbf{R} = \frac{1}{N}\mathbf{X}\mathbf{X}^{\text{T}} \tag{A9}$$

Since $L = N$, $\mathbf{X}$ becomes a square matrix. Moreover, since the $L$ bands are linearly independent to each other, $\mathbf{X}$ is reversible. Then the inversion of $\mathbf{R}$ can be expressed as:

$$\mathbf{R}^{-1} = \left(\frac{1}{N}\mathbf{X}\mathbf{X}^{\text{T}}\right)^{-1} = N\left(\mathbf{X}^{\text{T}}\right)^{-1}\mathbf{X}^{-1} \tag{A10}$$

Substitute (A10) into (1), the CEM detector can be rewritten as:

$$\mathbf{w}_{\text{CEM}} = c_{\text{CEM}}\mathbf{R}^{-1}\mathbf{d} = c_{\text{CEM}}N\left(\mathbf{X}^{\text{T}}\right)^{-1}\mathbf{X}^{-1}\mathbf{d} \tag{A11}$$

Then we can have the output, $\mathbf{y}$ as:

$$\mathbf{y} = \mathbf{X}^{\text{T}}\mathbf{w}_{\text{CEM}} = c_{\text{CEM}}N\mathbf{X}^{\text{T}}\left(\mathbf{X}^{\text{T}}\right)^{-1}\mathbf{X}^{-1}\mathbf{d} = c_{\text{CEM}}N\mathbf{X}^{-1}\mathbf{d} \tag{A12}$$

Then $\mathbf{d}$ can be expressed as:

$$\begin{aligned}
\mathbf{d} &= \frac{1}{c_{\text{CEM}}N}\mathbf{X}\mathbf{y} \\
&= \frac{1}{c_{\text{CEM}}N}\begin{bmatrix} \mathbf{x}_1 \ \mathbf{x}_2 \dots \mathbf{x}_{N_b} \ \mathbf{d}_1 \dots \mathbf{d}_{N_t-1} \ \mathbf{d} \end{bmatrix}\begin{bmatrix} y_1^b \ y_2^b \dots y_{N_b}^b \ y_1^t \dots y_{N_t-1}^t \ y_{N_t}^t \end{bmatrix}^{\text{T}} \\
&= \frac{1}{c_{\text{CEM}}N}\left(y_1^b\mathbf{x}_1 + y_2^b\mathbf{x}_2 + \cdots + y_{N_b}^b\mathbf{x}_{N_b}\right) + \frac{1}{c_{\text{CEM}}N}\left(y_1^t\mathbf{d}_1 + \cdots + y_{N_t-1}^t\mathbf{d}_{N_t-1} + y_{N_t}^t\mathbf{d}\right)
\end{aligned} \tag{A13}$$

Since $L = N$, and the bands are linearly independent, $\mathbf{x}_1, \mathbf{x}_2, \dots, \mathbf{d}_{N_b}, \mathbf{d}_1, \mathbf{d}_2, \dots, \mathbf{d}_{N_t-1}, \mathbf{d}$ are also linearly independent to each other. So,

$$\begin{cases} y_1^b = y_2^b = \cdots = y_{N_b}^b = y_1^t = \cdots = y_{N_t-1}^t = 0 \\ y_{N_t}^t = c_{\text{CEM}}N \end{cases} \tag{A14}$$

(A14) indicates that except the output of the selected target pixel, the output values of all the left pixels, including the other target pixels, are equal to 0.

Next, we will prove that the output of the selected target pixel $y_{N_t}^t$ is equal to 1. Substitute $\mathbf{R}^{-1}$ (A10) into $c_{\text{CEM}}$ (1), and one can obtain:

$$c_{\text{CEM}} = \frac{1}{\mathbf{d}^{\text{T}}\mathbf{R}^{-1}\mathbf{d}} = \frac{1}{N\mathbf{d}^{\text{T}}(\mathbf{X}^{\text{T}})^{-1}\mathbf{X}^{-1}\mathbf{d}} = \frac{1}{N(\mathbf{X}^{-1}\mathbf{d})^{\text{T}}\mathbf{X}^{-1}\mathbf{d}} \tag{A15}$$

Based on (A12), we can have:

$$\mathbf{X}^{-1}\mathbf{d} = \frac{1}{c_{\text{CEM}}N}\mathbf{y} \tag{A16}$$

Substitute (A14) and (A16) into (A15), and we can have:

$$c_{\text{CEM}} = \frac{(c_{\text{CEM}}N)^2}{N\mathbf{y}^{\text{T}}\mathbf{y}} = \frac{(c_{\text{CEM}}N)^2}{N\left(y_{N_t}^{\text{t}}\right)^{\text{T}}y_{N_t}^{\text{t}}} = \frac{(c_{\text{CEM}}N)^2}{N(c_{\text{CEM}}N)^2} = \frac{1}{N} \tag{A17}$$

Then substitute (A17) into (A14) and we can have $y_{N_t}^{\text{t}} = 1$. □

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
