# Peer review of "Hyperspectral Target Detection Methods Based on Statistical Information: The Key Problems and the Corresponding Strategies"

_remotesensing, doi:10.3390/rs15153835_

Round 1

Reviewer 1 Report

The authors have given the corresponding strategies for several common situations in the practical target detection applications based on the statistical characteristics of an image are always occupied a dominant position. The manuscript is complete, and the authors try to prove the progressiveness of the algorithm through experiments. However, there are some problems that need to be revised. The comments are as follows

1.      Please explain the innovative points of your manuscript, so that readers can better grasp the key points of the manuscript.

2.      Do the authors understand the definition of object detection in hyperspectral images? I think this manuscript is a bit off topic. Is it anomaly detection? Is it hyperspectral image classification?

3.      There are many obvious formatting errors in the manuscript, such as the labeling in Figure 2.

4.      The authors' introductions should be more detailed, as I believe they lack sufficient understanding of the latest information in hyperspectral image processing. Therefore, some of the latest methods need to be referenced, such as Multi-scale Receptive Fields: Graph Attention Neural Network, Multi-scale Receptive Fields: Graph Attention Neural Network, MultiReceptive Field: An Adaptive Path Aggregation Graph Neural Framework, Multi-feature Fusion: Graph Neural Network and CNN Combining. Additionally, I strongly disagree with the views of some authors, such as “When there are enough training samples, this type 96
of methods can usually obtain satisfactory target detection results. However, in actual 97
applications, it is difficult to meet this condition, especially when the target size is 98
small. In addition, this type of algorithm requires massive data and computational 99
conditions, and its computational complexity is also high.

5.      What is the basis for the authors' band selection? Can a simple experiment use a simple dataset to draw universal conclusions? I feel there is a big question about this.

6.      What are the criteria set by the authors? Why do these situations exist? In addition, the small goals mentioned by the authors are not actually small in the experiments conducted.

7.      The authors used a very old method to verify whether the feasibility of the proposed measures is correct, which requires further discussion. I have reservations about the conclusions drawn from the paper.

The  manuscript requires major revisions.

Language needs to be concise, and manuscript writing needs to be re improved.

Reviewer 2 Report

Please refer to the attachment for detailed suggestions.

Overall, the English language of the manuscript is good, and the author needs to check for spelling errors and tenses in the manuscript.

Round 2

Reviewer 1 Report

No more comments.